
# In-situ sounding of radiation flux profiles through the Arctic lower
# troposphere
Ralf Becker[1], Marion Maturilli[2], Rolf Philipona[3], Klaus Behrens[1]
[1]Deutscher Wetterdienst, Meteorologisches Observatrum Lindenberg/Mark,
Am Observatorium 12, D-15848 Tauche
[2]Alfred-Wegener-Institut, Helmholtz-Zentrum fuer Polar- und Meeresforschung
Telegrafenberg A45, D-14473 Potsdam
[3]MeteoSuisse, Chemin de l'Aerologie, CH-1530 Payerne
*Correspondence to:* Ralf.Becker@dwd.de
**Abstract**. In-situ profiles of all four net radiation components were obtained at Ny Ålesund/Svalbard (78.9° n,
11.9° e) in the time frame May 04-21, 2015. Measurements could be performed using adapted high quality
instrumentation classified as 'secondary standard' carried by a tethered balloon system. Balloon lifted
measurements of albedo under clear sky conditions demonstrate the altitude dependence of this parameter
over heterogeneous terrain. Depending on the surface composition within the sensor's footprint, the albedo
over predominantly snow covered surfaces was found to decrease to 53.4 % and 35.8 % compared to  73.1 %
and 78.8 % measured with near surface references, respectively. Albedo profiles show an all-sky maximum at
150 m above surface level, and an averaged vertical change rate of -2.1%/100m (clear sky) and -3.4%/100m
(overcast) above. Profiling of arctic low-level clouds reveals distinct vertical gradients in all radiation fluxes but
longwave upward. Observed radiative cooling at cloud top with heating rates of -53 to -84 K/d in subsequent
observations tend to be lower than suggested by 1-D simulations.

## 1. Introduction

Solar incoming radiation and the evoked shortwave and longwave interactions in the atmosphere and at the
Earth surface are the driving forces of our weather and climate. Particular importance for the Earth's energy
budget is attributed to clouds, as they reflect incoming solar radiation back out into space while holding
thermal radiation in the atmosphere (Ramanathan et al., 1989). Their actual radiative effect though depends
on the cloud temperature, phase, surface albedo and sun elevation. Especially in the Arctic, where solar
radiation with low elevation angles over the high albedo of snow and sea ice  encounters low level clouds, the
cloud net radiative effect  results in a warming of the surface through most of the year (Shupe and Intrieri,
2004). Climate forcing induced changes in cloudiness (Norris et al., 2016) may alter the atmospheric radiation
budget. Recent studies relying on high-quality ground-based radiation network data revealed that shortwave
downward radiation is subject to change through the decades (Wild, 2012). Accurate observations of the
components of the near surface radiation budget is therefore an essential task (IPCC, 2007). Such quality
ground-based measurements are performed in the context of the Baseline  Surface Radiation Network (BSRN,
Ohmura et al., 1998). The single components are quantified on a global scale based on satellite data with a
reasonable uncertainty, too. (Trenberth et al., 2009).

To fully understand cloud radiative effects and other radiative interaction in the atmospheric column, the
ground based data quality standard needs to be expanded into the free atmosphere. Measurement  equipment
mounted on masts and towers at different heights are used to investigate fluxes in the lower range of the
planetary boundary layer. For mid-latitude clear-sky and moderate wind conditions the radiative flux
divergence between 2 and 48 m above ground tends to be largest in the early evening hours (Sun et al., 2003).
To determine the radiative cooling at night Steeneveld et al. (2010) analyzed net longwave radiation at several
steps from 1.3 to 20 m above ground and found typical longwave heating rates of -1.8 $Kh^{-1}$ below 10 m and -0.5
$Kh^{-1}$ between 10 and 20 m height, respectively. To expand the ground operation to higher levels several
platforms carried by balloons were developed. There is a long history of vertical in-situ sounding of


meteorological parameters using tethered balloons and kites, starting already in the late 19th century. A
comprising review of early activities, developments and technical fundamentals is provided in Dubois (1961)(in
German). Duda and Stephens (1989) operated such a platform with standard meteorological instruments,
longwave and shortwave radiation sensors and a cloud droplet particle analyzer to determine microphysical
and radiative properties of stratocumulus clouds in a marine boundary layer. They compared measured vertical
irradiation profiles to calculated fluxes. Even though correspondence was quite good in the longwave range,
discrepancies in solar heating rates in the clouds appeared to be larger than expected. A lighter
instrumentation carrying short- and longwave sensing radiometers was developed by Alzheimer et al. (1993).
Their lifted platform was carrying short- and longwave sensing radiometers which were properly levelled
inflight by using accelerometers and a mechanical correction system. Siebert et al. (2003) developed a scientific
payload regarding standard meteorology and microphysical properties (aerosol particles, cloud droplets), both
suspended below a tethered balloon. The focus of this experiment was to determine turbulent fluctuations
with high temporal resolution. A system for sounding radiation fluxes through the whole troposphere up to the
lower stratosphere was recently introduced by Philipona et al. (2012) and Kräuchi and Philipona (2016). While
the development of ground-based and satellite-based remote sensing techniques utilizing active (radar, lidar
and sonic) and passive approaches advances, there is also a growing need for high quality in-situ sensing for
validation, ground truthing, and high resolution process studies.
In this study we discuss results of profile measurements of the net radiation components obtained under clear-
sky and cloudy conditions with a tethered balloon in an Arctic environment. Section 2 is dedicated to the
radiation sensors, the sounding set-up, and the interpretation and correction of data. Special emphasis is given
to the handling of tilting during flight and its correction. Section 3 introduces the radiation flux profiling
campaign conducted at Ny-Ålesund, Svalbard. One focus is set on land surface characterization with respect to
the observation height under both clear sky and overcast conditions. The second focus is set on radiation flux
profiling in the presence of clouds. Here a case study probing a dissolving Arctic stratus cloud is discussed.
Some results of comparisons to simulated fluxes and cooling effects associated with clouds are given in section
4. A summary of the study outcome and an outlook on potential further activities is provided in section 5.

## 2. Instruments and Methods
### 2.1 Radiation sensors and sounding set-up
Measurements of solar and terrestrial radiation need to be performed under certain conditions to ensure best
possible quality and representativeness. A first and important point is the selection of sensors. The selected
sensors for our study have a high quality that is achieved within a 3 % margin to reference of 95 % of the
readings in hourly total and 2% in daily totals (WMO, 2008). For the purpose of net radiation observations in
the lower troposphere a dedicated sonde was developed, consisting of two paired upward and downward
looking shortwave and longwave instruments, respectively. For the shortwave (solar) range, CM22
pyranometers by Kipp&Zonen are applied for the sake of their reduced sensitivity to fast changes of ambient
air temperature. Measurements of longwave (terrestrial) radiation fluxes are obtained with CG4 pyrgeometers
by Kipp&Zonen. Both, short- and longwave instruments are categorized as secondary standard. Calibration is
performed in the field in a yearly cycle at the National radiation measurement central facility located at the
Lindenberg Meteorological Observatory of Deutscher Wetterdienst and therefore linked to world radiometric
reference (WRR) via the local standard group. Tracking the calibration factors of all sensors before and after
construction of the sonde revealed no distinct change in the characteristics of the sensor by means of altered
factors.  The instrument bodies were modified to allow the mounting as upward and downward oriented pairs.
To save weight no radiation shields were used. In general, hardware weight was saved wherever feasible to
increase the free buoyancy. In contrast to near surface observations no ventilation was mounted assuming an
adequate air stream present during all flights. The instruments are mounted and levelled on a glass fiber pole
attached in a flexible holder and to be hooked in the tethered line. A wind vane on the back enables horizontal
orientation of the sonde in the air stream. Figure 1 is showing the instrumentation. For the data acquisition two
commercial light-weight loggers manufactured by Driesen&Kern are used. In total four loggers are needed to
store signals, instrument temperatures and tilting angles.



The storage interval can be chosen from 1 second to 1 minute. Probing the free atmosphere requires to
operate the equipment within the range of meteorological parameters proposed by the manufacturer. This is
particularly important for the ambient air temperature range. Earlier CM11 pyranometers had to be operated
within a range from -10° to 40°C. For the CM22 pyranometer as well as for the CG4 pyrgeometer the range of
low temperature sensitivity is extended down to -20°C. Another important number is the stability of the signal
in case of strong temperature lapse rates. Usually it is described that a gradient of 5 $Kh^{-1}$ maximum will keep
the so-called Zero-offset B below 2 $Wm^{-2}$, which is quite a low effect. On the other hand it is expected to face
temperature gradients of $5 - 10\ Kh^{-1}$ in flight. To minimize this kind of temperature adaption effects the sensors
need to be exposed to ambient conditions at least 30 minutes before ascent. A payload of 1060 gr is resulting
plus about 250 gr for cables, plugs, loggers and housing.

### 2.2 Carrier and other sensors

122  Soundings at Svalbard were conducted using a Vaisala TT12 tether sonde system. It comprises the following
123  components:
- a streamlined tethered balloon of TTB329 series with a length of 5.2 m, a maximum diameter of 2.3 m and a volume of 9 $m^3$, made of 0.08 mm polyurethane
- an electrical winch TTW111 with automatic engine shutdown including about 2000 m tether line
- a DigiCora III MW21 receiving system comprising laptop, receiver, antenna and the software
- several tether sondes of type TTS111 for in situ measurements of meteorological standard parameters

Wind, temperature and humidity data are measured with 1 Hz and transferred to ground via telemetry. Thus,
the meteorological receiving system and the radiation sensor recording need to be synchronized at least before
beginning of the measurements or time strings need to be corrected afterwards. Due to the slower response of
the radiation sensors synchronization is no critical issue here.
Tethered ballooning allows direct in-situ measurements of the selected quantities. The usability of this
technique however, is restricted by meteorological conditions, i.e. it requires low winds in the sensed air
column (best: less than 5 $ms^{-1}$, endurable: less than 10 $ms^{-1}$) and no precipitation. Taking into account an
averaged ascending/descending rate of 2 $ms^{-1}$ a steady-state radiation field should be given to minimize mixing
of temporal and spatial (vertical) change. Maximum sensing height in this campaign was 2 km above ground as
restricted by the local flight control authority. The equipment allows for several consecutive hours of
measurements at dedicated heights.

### 2.3 Quantification and correction of tilting

Tilting of the sonde may be a potential source of error, mainly regarding the shortwave downward flux.
Assuming a maximum tilting of the sonde of 10 degrees with respect to the observed sun zenith angle and
recognizing the lack of a direct component in all upward fluxes as well as longwave downward these three
components are no subject of further discussion here. Concerning solar incoming radiation the tilting can be
described as a disturbance in the sun zenith angle between the horizontal plane (true sun zenith) and the
sonde (observed). Such disturbances may be characterized as prevailing systematic deviations due to
uncertainties that can hardly be avoided when levelling the sonde before start. First rough estimates of the
impact of a systematic tilting revealed high potential errors that need to be corrected accordingly. It needs to
be treated more seriously than fast signal variations caused by turbulent flow.
Based on measured data, Figure 2 shows exemplary that an error of 5 degrees in solar zenith angle would
produce an error of 96 $Wm^{-2}$ in shortwave downward radiation, i.e. almost 25 % of the total signal. To
determine the observed horizon, the three Eulerian angles need to be measured. Here, the roll and pitch angles
are obtained with a two-axis inclinometer based on MEMS technology, fabricated by Althen Mess- und
Sensortechnik GmbH, Germany. It can be operated over a wide range of ambient temperatures (-25 to +85°C)
and is mounted on the CM22 pyranometer pair. The yaw angle is determined as the difference between wind
direction - measured by the wind vane – and apparent azimuth of the sun (calculated). Using this input a
transformation of coordinates can be applied, yielding to the individual sun zenith at the time of measurement.
In cloudy and overcast condition the misalignment error is negligible.





The correction scheme follows the idea that deviating the sensor from its intended horizontal alignment may
lead to a loss or a yield in direct downward radiation. This deviation can be quantified by a function $SW_{down}$ =
f(theta) delineated by a polynomial fit of second order of collocated near-surface measurements of shortwave
downward radiation. It is assumed that the amount of other atmospheric constituents, like ozone, aerosol load
and water vapor, remains constant over the time range of investigation, usually one to several hours. To check
whether this function, based on ground measured data, represents the conditions not only close to the surface
but in all of the lower troposphere, we refer to measured irradiation data. Examples for the closely located
sites Garmisch-Partenkirchen (GAP, 704 m a.s.l., 47.49° north, 11.10° east) and Zugspitze (ZUG, 2962 m a.s.l.,
47.42° north, 10.98° east) are displayed in Figure 3. Both sites have only 10 km linear distance. An almost clear-
sky day with high insolation was chosen. Fitting procedure comprises of 2 to 3 steps depending on the number
and deviation of outliers, that can be caused by the passage of cirrus clouds or small cumulus clouds. These
outliers are identified and excluded by the fitting algorithm. Despite of the high difference in altitude both
fitting functions are quite similar. The total effect of potential misalignment on shortwave downward fluxes for
both sites agrees at given discrete angles as displayed in figure 4. With respect to the usual range of solar
zenith angles in Central Europe (not less than 25 degrees) the error can be up to 150 Wm$^{-2}$. But for the proper
correction of sonde data only the difference of the correction term between valley and mountain site is
relevant. It never exceeds 10 Wm$^{-2}$ and is lower than 5 Wm$^{-2}$ at tilts of 3° and 6°. The three technical
components involved here - inclinometer, meteorological wind sensor, irradiation sensor - differ in response
times. Thus, fast changes in radiation as may occur at the ascent and descent of the sonde or in turbulent or
gusty conditions may be not be observed. Therefore tilting correction is only applied to situations with low
variability in yaw, roll and pitch angles. In case of cloud profiling the correction is applied above the cloud top if
low wind variability is given.

## 2.4 Calculations

The net radiation $F_{net,i}$ is defined as the difference of downward and upward fluxes, thus for the net flux at
atmospheric level i follows

$$F_{net,i} = SW_{down,i} - SW_{up,i} + LW_{down,i} - LW_{up,i} \tag{1}$$

where $SW_{down,i}$ is the downward shortwave (solar) irradiance, comprising direct and diffuse radiation at level i,
$SW_{up,i}$ the upward shortwave irradiance, $LW_{down,i}$ the longwave (terrestrial) irradiance emitted by the
atmosphere and $LW_{up,i}$ the upward longwave irradiance stemming from surface and atmospheric column
under investigation. To get  the net terrestrial irradiance $F_{LWnet,i}$ the solar components are neglected:

$$F_{LWnet,i} = LW_{down,i} - LW_{up,i} \tag{2}$$

Computations are performed based on a vertical resolution of 50 to 100 m. To calculate the radiative
heating/cooling rates (HR) at level i we use

$$HR_i = 1 / (c_p * \rho(z_i)) * \delta F_{Net,i} / \delta z_i \tag{3}$$

where $c_p$ is the specific heat capacity of air, $\rho(z_i)$ the air density, being a function of altitude $z_i$, and $\delta F_{Net,i} / \delta z_i$
the net radiation flux divergence

Analogous, a longwave heating rate is calculated taking into account the longwave net radiation flux divergence
$\delta F_{LWNet,i} / \delta z_i$ , only. For simulation of fluxes the freely available radiative transfer package *Streamer* (Key and
Schwaiger [1998]) was utilized. *Streamer* can be applied to simulate broadband fluxes for a wide variety of
atmospheric states. User input of atmospheric and trace gas profiles is feasible. A great number of cloud ice
particle models and surface characteristics is provided. In our study, we constrain the atmospheric state in the
model by the following measurements:

- profiles of temperature and humidity: tethered balloon based soundings or alternatively routine
  radiosoundings (available daily at 11 UT)
- ozone: column amount from the ozone monitoring instrument (OMI), available at
  https://aura.gsfc.nasa.gov





• AOD at 600 nm: local readings from SP1A sun photometers
• surface temperature $T_{surf}$: calculated from BSRN measurement of longwave upward radiation readings
at 1.30 m above ground using
$$T_{surf} = \sqrt[4]{\frac{LWU}{\varepsilon\_surf * \sigma}}$$   (4)
where LWU is longwave upward radiation measured close to ground, $\varepsilon\_surf$ the surface emissivity and $\sigma$ the
Stefan-Boltzmann-constant.
• surface emissivity of snow with respect to age and estimated grain size of snow cover is set according
to the findings of Hori et al. (2006)
• cloud base height from ceilometer (Maturilli, 2016)
Temperature at the cloud base is given by the sounding itself. Geometrical cloud thickness is detected in the
radiation data in case of fully passing a cloud. Concerning the microphysical parametrization of the clouds –
liquid water content and effective radii of the droplets - mean summertime Arctic clouds characteristics
according to the review in Curry and Ebert (1992) were used.
## 3. Observations
### 3.1 Site characteristics Ny Ålesund (78.9°N; 11.9°E)
Ny Ålesund is an international research community located at the Kongsfjorden on the west coast of Svalbard
(Spitsbergen). Here, the Alfred Wegener Institute operates a variety of atmospheric measurements including
basic surface meteorology (Maturilli et al., 2013), daily radiosoundings (Maturilli and Kayser, 2016), weekly
ozone soundings, AOD retrieval using sun photometer, and radiation measurements contributing to the BSRN
(Maturilli et al., 2015). Campaign operated soundings using tethered balloons as a carrier at  Ny Ålesund are
restricted in timing  by air traffic requirements and are generally limited to a maximum height of 2 km above
ground. The tethered balloon campaign in May 2015 was core part of the project "Vertical profile of the net
radiation balance" that was initiated to find characteristic clear sky profiles of net radiation as well as to get a
snapshot of the radiative profile  under typical Arctic boundary layer cloud conditions.
Vertical soundings by tethered balloon take time, about 20 minutes for a profile reaching 1000 m above
ground. Therefore, stationarity is required during recording of a profile. Consequently shortwave flux profiles
for the investigation of albedo are rejected in case of changing cloudiness.
### 3.2 Albedo
Albedo as an important surface property is subject to changes in mid latitudes as well as in the Arctic
throughout the year by the growing cycle of vegetation and/or the existence and varying persistence of snow
cover, respectively. It is defined as the ratio of outgoing and incoming shortwave fluxes and is conventionally
measured near the surface. For the Artic site at Svalbard snow disappearance can be characterized as a sharp
transition phase from high (0.8) to low albedo values (0.1), usually taking place between end of May and
beginning of July. Recent investigations showed a tendency towards an earlier start of melting referring to the
21-year observation period (Maturilli et al., 2015).
In flat homogeneous terrain shortwave upward fluxes from the surface are only modified by radiative transfer
processes. It can be expected to be different over heterogeneous terrain as the footprint of the sensor enlarges
and the observed composition of the surroundings changes with height. In Ny-Ålesund, the surface based $SW_{up}$
instrumentation has only the (snow-covered) surface in its downward looking hemispherical view. From the
higher elevations of the tethered balloon platform, the enlarged hemispherical view will include some
infrastructure of the village (houses, roads), the dark surface of the open fjord water, and effects of the inclined
surface of the mountains. Moreover, the clear-sky upward shortwave fluxes are strongly linked to sun zenith
angle at the time of measurements. This causes a wider range of absolute fluxes in clear-sky compared to
overcast conditions. In case of uniform cloudiness measurements in the clouds are left out here.



The measured shortwave upward flux shown in Figure 5 is the quantity describing the surface property (and
the atmospheric layer between surface and sensor) whereas the incoming flux is just needed to normalize the
result. Here, for the direct sun a simulated shortwave downward flux is used to apply this normalization. In
absence of insolation and therefore absence of directional error the shortwave downward measurements are
used. The profiles of albedo, grouped to 'clear sky' (5 profiles) and 'overcast' (7 profiles) in Figure 6 show a
maximum at 150 to 250 m asl ranging from 63 to 74 % with greater values to lower sun elevation in clear sky
conditions. The albedo profiles at overcast conditions tend to show 4 to 8 % higher values at all height levels.
Albedo decreases with height by -2.1%/100m in clear sky conditions over a range of 150 m to 1150 m above
sea level. In overcast situation a rate of -3.4%/100m is observed (from 150 to 750 m asl). Generally, a similar
characteristics of the profiles can be found in all soundings: a strong gradient close to ground leads to the
maximum albedo at 150 to 250 m followed by decreasing values with height.
Clear sky soundings with several hours observation time almost undisturbed by clouds at fixed altitudes were
performed on May 08th (cirrus clouds at 6 to 7 km base height at 1500, 1930 and 2045 UT according to
ceilometer record) and on May 13th, 2015 (sudden end at 1840 UT due to a stratus cloud deck based at 600
m). The meteorological conditions in the local boundary layer at dedicated observation levels were quite
similar: temperatures at about -7°C on May 08th (at 882 m asl) and about -8°C on May 13th (at 494 m asl) with
winds below 5 ms$^{-1}$ from east-southeast (Figures 7 and 8), respectively. Relative humidity was almost constant
at about 51% on May 08 but showed an increase from 50 to 75% on May 13th. The corresponding time series
of the albedo are given in Figure 9. On May 8th we get 35.8 ± 1.9 %, whereas 53.4 ± 1.8 % is observed 5 days
later. The surface-based observations do vary only slightly on a remarkably higher absolute level. Averages
matching to the observation times of the sonde are 78.8 ± 1.7 % and 73.1 ± 1.4 %, respectively. The large
discrepancies between near surface and lifted observation is caused by change of the scenery caught by the
sensor.  Furthermore ideally, a tethered balloon sounding will lift up the sonde vertically and the perpendicular
will be above the winch. In reality even when using streamlined balloons a horizontal displacement occurs
depending mostly on wind speed and free buoyancy. Referring to the case study given here we notice almost
identical wind direction on both days but wind speed is more than doubled on May 08: 2.6 ms$^{-1}$ versus 0.8 ms$^{-1}$,
respectively. Due to the lack of exact spatial information (no GPS onboard) a horizontal drift of 200 m towards
northwest is assumed. It corresponds to an averaged attack angle of the rope of 77.5°, which is in line with
onsite visual observation. According to this the perpendicular of the sonde on May 08 was situated at the
coastline, separating bright, prevailing snowy land surface and dark ice-free sea water surface. Hence,  the
upward shortwave flux was composed by snow over land and to a wider extent by ice free sea surface.

314       3.3  Radiative fluxes under clear sky and cloudy conditions

Atmospheric longwave downward radiation usually shows a well-defined decrease with increasing height in a
well-mixed boundary layer due to the adiabatic temperature gradient. Exceptions in case of thermal inversions
are a common feature in the Arctic. An example for a thermal inversion was observed on May 13th at 1239 UT
(May13 1 in Figure 10). Here, a thermal inversion layer at 750 m a.s.l. was detected, which is obvious in the
profile of potential temperature above 750 m. The radiative effect results in an increase of longwave
downward radiation of 7.4 Wm$^{-2}$ between 650 and 750 m. It corresponds to an increase of longwave upward
radiation of 6.0 Wm$^{-2}$ at the same level. Due to its almost counterbalancing in net effect the thermal inversion
cannot be detected in the net longwave radiation profiles. Still, the importance of thermal inversion layers for
longwave downward radiative heating within the atmospheric column and its potential contribution to the
bottom-amplified warming of the Arctic atmosphere should be considered. Obviously, the effect of clouds on
the longwave radiation is more pronounced. Even though net longwave flux is still negative at overcast
conditions the radiative effect is about +80 Wm$^{-2}$ close to ground and +90 Wm$^{-2}$ at 800 m a.s.l. (compared to
clear-sky conditions, see figure 11).
Low level clouds are a worthwhile subject of investigation with tethered balloon borne equipment. Taking into
account restrictions of the equipment plus limitations by the flight control authority we were awaiting stratus
or stratocumulus clouds with cloud base height at 500 to 1300 meters and a vertical thickness of 500 meters
maximum. These types of clouds - supported by cooling surface - occur quite often in the Arctic region (Tsay et





al., 1989). On two days (May 11th and 12th, 2015) cloud flights could be performed. A total of 4 consecutive
profiles taken on May 12th, describing the transition phase from overcast to partly cloudy conditions are
discussed here . From 12 a.m. to 3 p.m. the BSRN station readings of shortwave downward radiation show a
large variability in the signal ranging from 250 to 400 Wm$^{-2}$. But the cloud deck is not fully disappearing and the
direct normalized shortwave downward radiation never exceeds 100 Wm$^{-2}$. The process could not be followed
further due to an advancing lower level cloud layer with 500 m base height. Table 1 provides an overview of
the retrieved cloud characteristics. The cloud base height is provided by local ceilometer measurements
(Maturilli, 2016). Temperature at the cloud base is identified from the in-situ radiosounding at the respective
height.  A local temperature minimum was detected at 1400 m (  -14°C). The air at 1.5 to 1.6 km was 2 K
warmer, and the lapse rate above was weaker than measured below the cloud (Figure 12). Conditions can be
characterized to have a well-mixed boundary layer below the clouds with relative humidity between 60 and 80
% and distinctively drier air above the clouds. Therefore, the drop of relative humidity to 20 % is taken as a
marker for the cloud top which varies between 1.4 and 1.5 km. Wind was showing only weak shear in velocity
with 4 to 5 ms$^{-1}$ at cloud level and 1 to 2 ms$^{-1}$ from southeast both in the boundary layer and above the cloud.
During measurements on May 12th the cloud layer commenced to dissolve. Solar fluxes show a distinct
increase of about 200 Wm$^{-2}$ when passing the cloud layer. Terrestrial downward radiation drops down to 130 to
140 Wm$^{-2}$, originating from the cloud free troposphere. In contrast, the drop of the longwave upward
component is only about half (Figure 13).
## 4. Discussion
### 4.1 Heating rates
Most of the variability in the net flux divergence comes from the shortwave downward radiation, here
composed of two effects: the increasing of insolation when passing the cloud and the observation of varying
solar flux below the cloud due to inhomogeneity in the cloud deck. The latter is identified as artificial: a change
in time pretends a change in space. The change in shortwave upward radiation peaks just above the cloud top.
No signal can be seen here at the cloud base (Figure 13).
Longwave downward radiation is decreasing rapidly when passing the cloud top with maximum rates
of -2 Wm$^{-2}$/m. With advancing time, the rates we obtained for SW$_{up}$ and LW$_{down}$ were lower due to the partly
dissolution of the cloud deck. The longwave upward component was always below 0.5 Wm$^{-2}$/m. In the albedo
profile, the passing through the cloud is less sharp pronounced. We observe an increase from 62 to 72% below
the cloud (except near surface feature mentioned earlier in this text) to 72 to 80 % above. In fact, the
shortwave net excess is only about 40 Wm$^{-2}$ due to the increased reflectance at cloud top. The total net effect
on the other hand is distinctly negative (-50 Wm$^{-2}$) caused by a sharp negative gradient in longwave downward
radiation above the cloud top while longwave upward radiation only slightly decreases. Local heating or cooling
is driven by the net flux divergence and thus by the vertical gradient of the net radiation. The measured heating
rates are shown in Figure 14. For the last profile at around 1400 UT the heating rate is left out intentionally due
to changes in cloudiness during the profiling. At all observation times, a radiative cooling at the cloud top with
rates up to 62 Kh$^{-1}$ was observed. It effects in a thermal inversion in the height range of 1400 - 1600 m with an
increase of temperature of 2 K. A shortwave excess causing a local warming can be found in the first profile at
1.5 km. Possibly a peak in solar incoming radiation related to side reflections due to cloud morphology is
responsible, whereas the terrestrial net radiation remains below zero. We have to assume that the cloud top
was no flat surface at observation time. Above the surface we observe the warming due to village
infrastructure (snow free roads, buildings) - an effect that would have been missed if observations could have
been performed on completely snow covered unobstructed field.




## 4.2 Model Comparison

| Mean time | Cloud base (cbh, km) | Temp. at cbh (K) | Cloud thick-ness (km) | Heating rate at cbh, observed ($Kd^{-1}$) | Heating rate at cbh, calculated ($Kd^{-1}$) | Heating rate at cloud top, observed ($Kd^{-1}$) | Heating rate at cloud top, calculated ($Kd^{-1}$) | Cloud radiative effect (short-wave, $Wm^{-2}$) | Cloud radiative effect (long-wave, $Wm^{-2}$) |
|---|---|---|---|---|---|---|---|---|---|
| 12:15 | 1.15 | 261.0 | 0.33 | +5.9 | +15.9 | -62.1 | -87.6 | -71.7 | +79.1 |
| 12:45 | 1.15 | 261.0 | 0.33 | +7.4 | +15.7 | -58.6 | -85.6 | -70.5 | +77.8 |
| 13:30 | 1.15 | 261.2 | 0.34 | -2.9 | +15.7 | -53.3 | -116.3 | -63.6 | +79.6 |
| 14:00 | 0.93 | 263.0 | 0.49 | +0.2 | +14.9 | -84.3 | -65.9 | -62.0 | +81.9 |

Table 1: cloud characteristics on May 12th, cloud base height (cbh) and geometrical thickness given in km, temperature at cloud base given in K, simulated and observed heating rates at cloud base and cloud top provided in $Kd^{-1}$, cloud radiative effect for shortwave and longwave components at surface level is given in $Wm^{-2}$

In the Streamer model, the atmospheric state in the lower troposphere is described by the sounding itself (up to the maximum height 1.575 km and 1.975 km). For the upper levels the routine radiosounding performed at 1100 UT is concatenated. To get a realistic picture of the cloud in the model some parameters need to be sampled. Cloud base height was derived from the local ceilometer (Maturilli, 2016). In the beginning of the observation sequence we get a clear and almost constant signal. Later on, an initially broken stratus layer has advected and there was more variability in the base height of the upper clouds. For 1400 UT only one shot can be employed. Temperature at cloud base is the sounded temperature at that height. Thickness of the cloud layer is derived using the additionally inferred cloud top height. It is the level when temperature lapse rate gets positive, i.e. the bottom of the temperature inversion. That level coincides very well to the lapse in relative humidity except for the profile around 1400 UT: here obviously a slack in the cloud is observed and the cloud is penetrated 60 meters lower. For the comparison of observed and simulated data we set a focus on cloud base and cloud top. Results of the Streamer model indicate clear positive heating rates when intruding the cloud base. For flight 1 around 1215 UT a decrease of 23 $Wm^{-2}$ in $LW_{up}$ is simulated, while $LW_{down}$ remains almost constant. The model provides similar results for the other flights as well. The corresponding heating rates of almost constantly 15 $Kd^{-1}$ are to a certain degree confirmed by the first two measurements, but not by the 1330 and 1400 UT profiles. The observations reveal a weaker gradient in terrestrial upward flux and therefore a lower heating rate. In contradiction, the cloud top is much more pronounced in radiation data, both in the model and in the observations. Considering that cloud top was more uniform in the beginning of the observation sequence implying a better representation in the model, we see an overestimation of the cloud cooling effect by about 41% (1215 UT) and 46% (1245 UT). Obviously, these are only first results on vertical radiation measurements within clouds. A profound study of the cloud radiative effects requires detailed knowledge on macro- and microphysical characteristics of the respective cloud, e.g. cloud optical depth, cloud effective radii, particle phase, and liquid water path.

## 5. Conclusions and outlook

In this study we present solar and terrestrial radiation flux profile observations in an Arctic environment. We focused on radiation features in the lower troposphere under clear-sky and overcast conditions. Vertical profiles of albedo reveal a strong dependence on the height of observation related to the heterogenous surface within the sensors' field of view. In the mountainous fjord surrounding of the Ny-Ålesund site, the dark surface of the open fjord water affected the observations over the snow covered land surface above 150m height. Above that level, a rate of -2.1%/100m was derived as typical for the local combination of open sea plus snow covered land surface. Observations of that kind have the potential to provide a broader view on land surface characteristics with special attention on linking near-surface measurements and satellite retrievals. Time series of albedo in fixed heights further demonstrate the relative robustness of net radiation observations using a tethered balloon platform. The measurement set-up also proofed its high potential to observe the longwave radiative effect of thermal inversions under clear sky conditions and to retrieve the radiative effect of clouds. A sequence of measurements in and above Arctic stratus clouds revealed a weak warming at cloud base (less than 10 $Kd^{-1}$) and a stronger cooling at the cloud top (53.3 $Kd^{-1}$ to 84.3 $Kd^{-1}$). The measurements are found



to be in reasonable agreement with model results based on the macro-physical characterization of the
individual cloud. Future combination with microphysical cloud in-situ measurements are expected to provide
further insight to irradiation features related to low level clouds at various latitudes.
## Acknowledgements

We kindly thank AWI for the opportunity to perform tethered balloon soundings at the AWIPEV research base
in Ny-Alesund, Svalbard. Thanks to the local flight control for supporting our issues.  The sounding procedures
were strongly supported by Jürgen Graeser, Ingo Beninga,  Rene Bürgi, and the AWIPEV station staff. We
further thank Christoph Ritter for providing AOD data. Special thanks to technical staff Jörg Karpinsky and
Steffen Gross at Meteorological Observatory  Lindenberg for adapting  the sensors, setting up the data
acquisition and supporting the improvement  of all handling aspects.

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

WMO, 2008. Guide to meteorological measurements instruments and methods of observation. Technical
report, World Meteorological Organization, 7th edition.



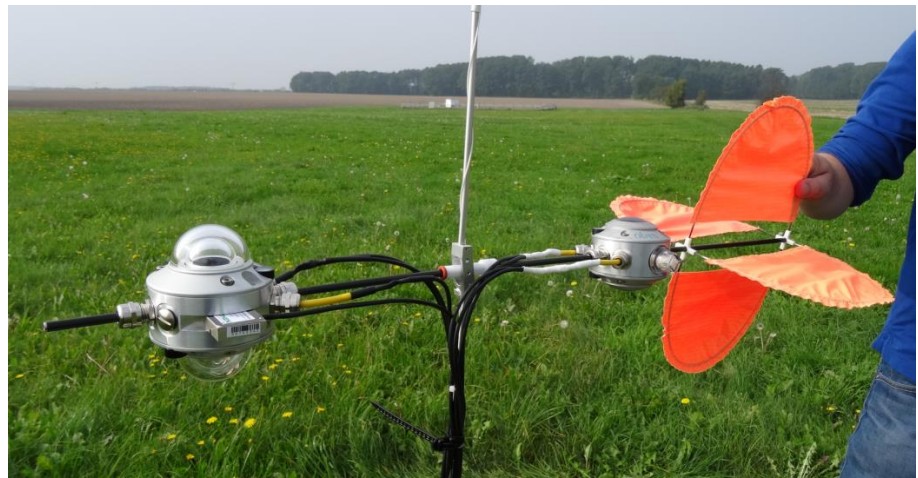


Figure 1: Instrumental setup of the net radiation sonde. A pair of CM22 and a fixed mounted 2-axis
inclinometer to the left, a pair of CG4 plus wind vane on the right hand side. A small box logger, housing the
loggers, is mounted about 1.5 m below the sonde.

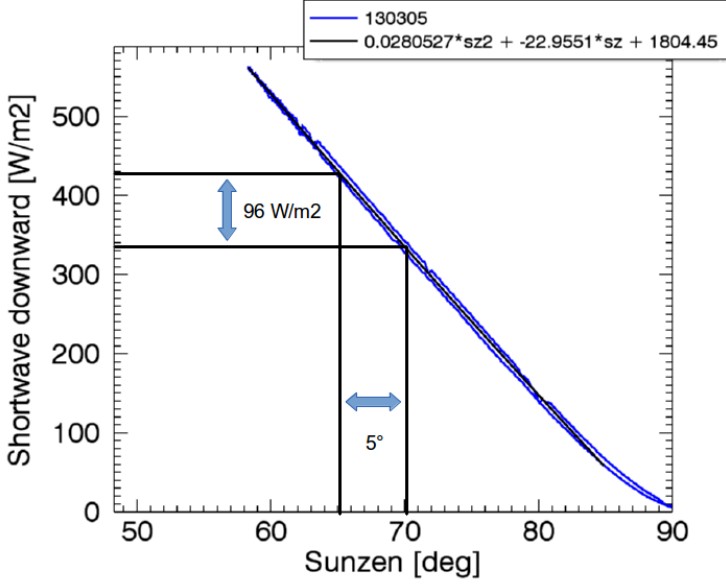

Figure 2: Shortwave downward radiation measured in Lindenberg/Germany on March 05, 2013, as a function
of sun zenith angle and polynomial fit of 2nd order.



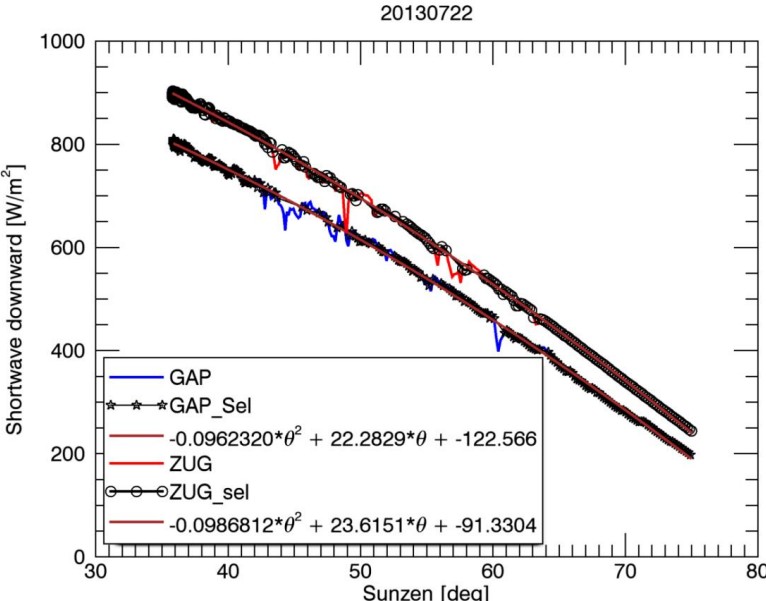

Figure 3: Data fitting to skip cloud contaminated phases, for the sites Garmisch-Partenkirchen and Zugspitze on
July 22, 2013.

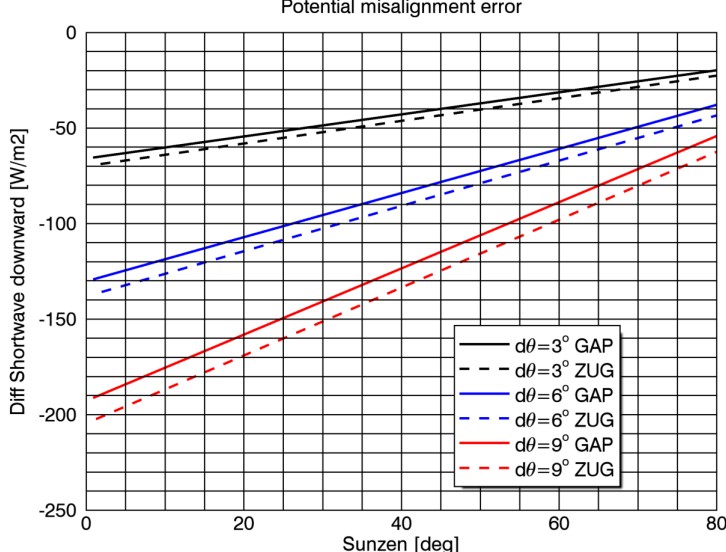

Figure 4: Quantification of the effect of misalignment on shortwave downward radiation measurements,
calculated for the sites GAP and ZUG, assuming 3 typical tilting angles.






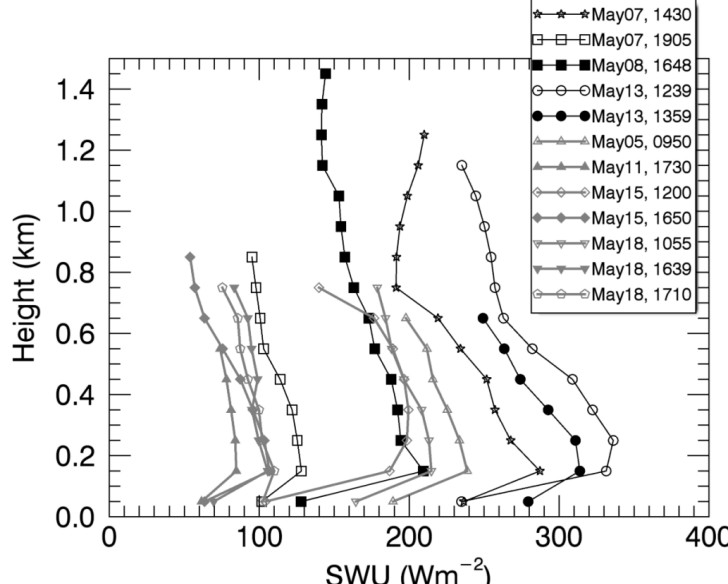

Figure 5: Profiles of shortwave upward fluxes above Ny Ålesund taken at clear sky conditions (black curves) and
at overcast conditions (grey).

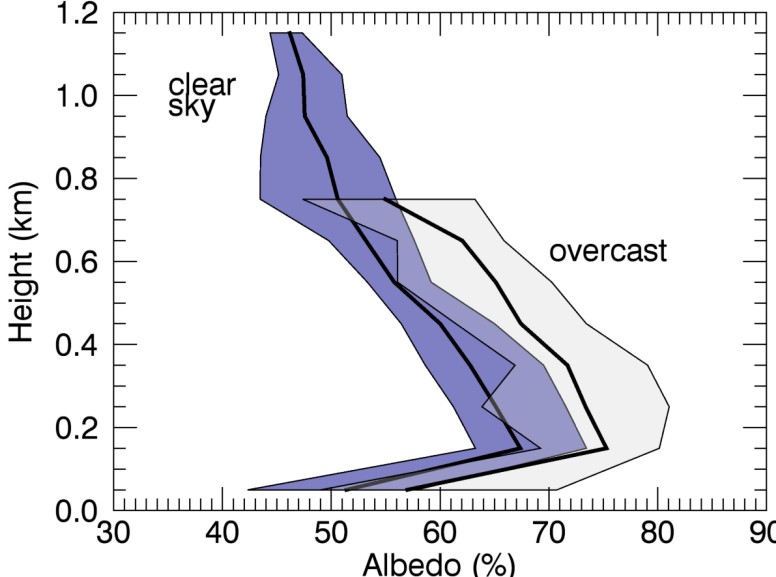






Figure 6: Averaged profiles of albedo and range, separated into the categories clear-sky (blue) and overcast
(light grey).

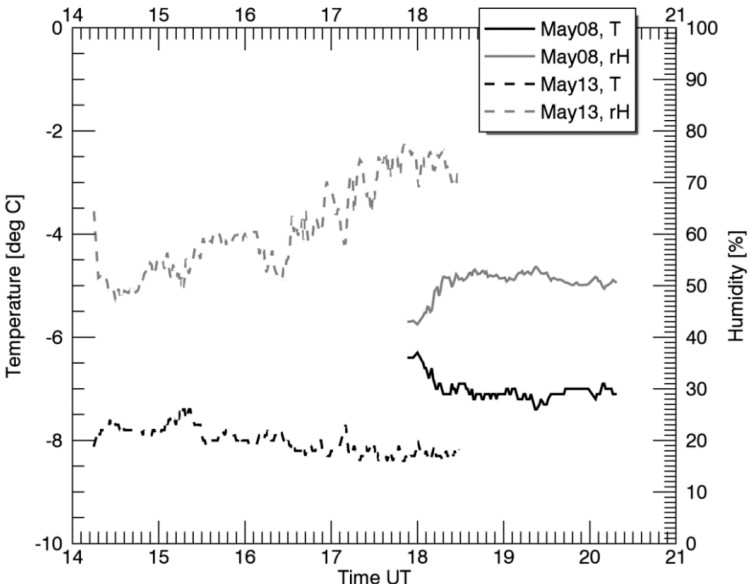

Figure 7: Timeseries of temperature and relative humidity on May 08, 2015, at 881.6 m asl ($\sigma$ = 8.5 m) and on
May 13 at 493.9 m ($\sigma$= 4.4 m), respectively.

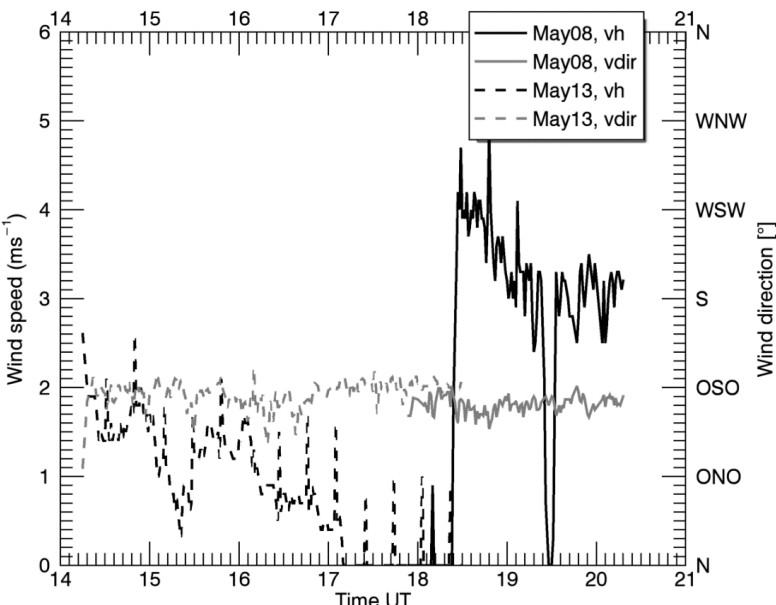






Figure 8: Timeseries of wind velocity and direction on May 08, 2015, at 881.6 m a.s.l. ($\sigma$ = 8.5 m) and on May 13
at 493.9 m ($\sigma$ = 4.4 m), respectively.

Figure 9: Clear sky albedo time series on May 08, 2015, at 881.6 m a.s.l. ($\sigma$ = 8.5 m) and on May 13 at 493.9 m
($\sigma$ = 4.4 m) versus near surface readings.

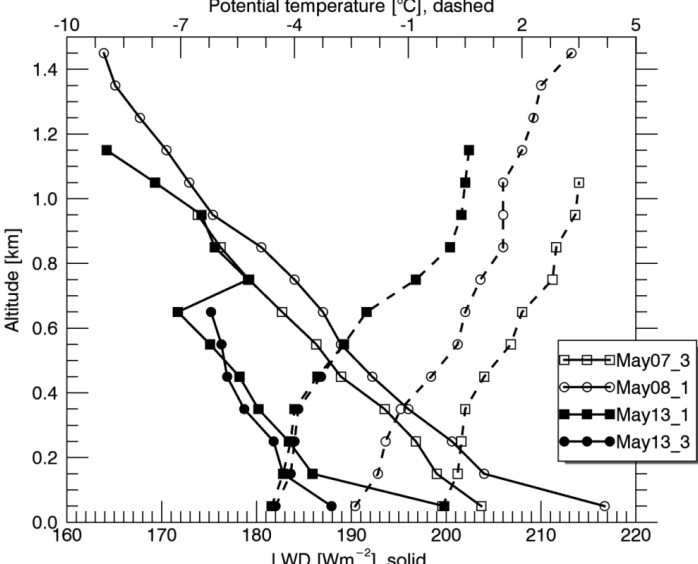

Figure 10: Profiles of longwave downward fluxes and potential temperature taken on May 07 to 13 above
Ny-Ålesund.






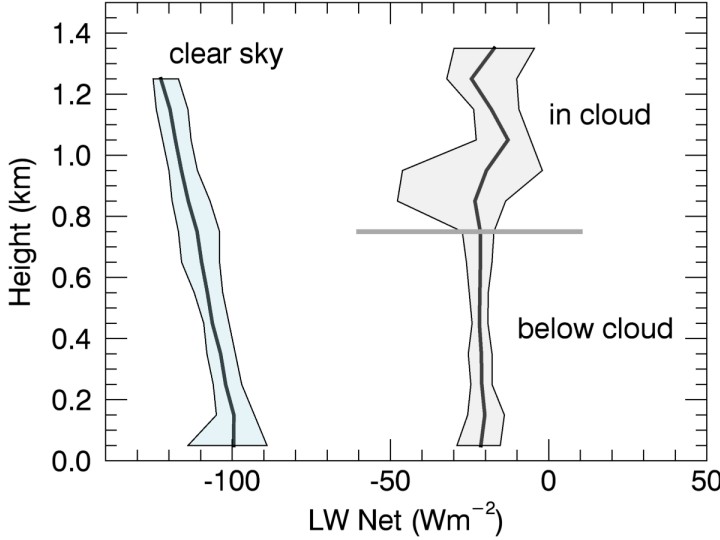

Figure 11: Net longwave fluxes taken above Ny-Ålesund for clear and overcast conditions

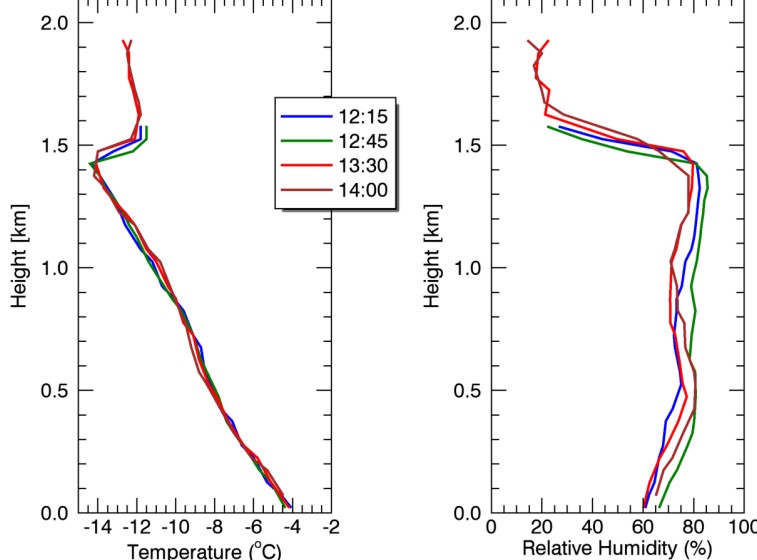




Figure 12: Profiles of temperature and humidity recorded during 4 cloud flights on May 12, 2015, at
Ny-Ålesund.

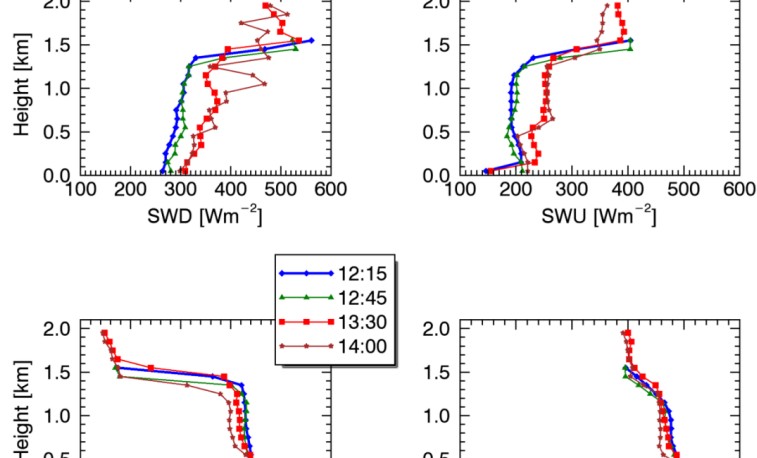

Figure 13: Radiation flux profiles recorded during 4 cloud flights on May 12, 2015, at Ny-Ålesund.





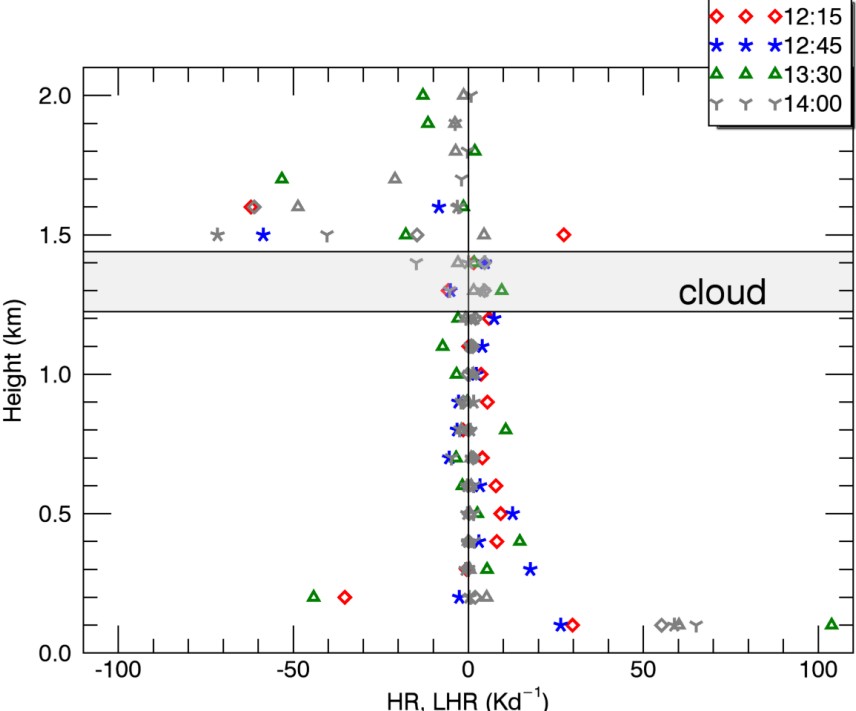

Figure 14: Profiles of heating rates retrieved for cloud flights on May 12, 2015. At 1400 only longwave heating
rate is provided. Numbers for cloud base (1.225 km) and cloud top height (1.44 km) are averages over all
profiles.