# Peer review of "Atmos. Meas. Tech. Discuss., https://doi.org/10.5194/amt-2018-173 Manuscript under review for journal Atmos. Meas. Tech. Discussion started: 14 June 2018"

_Atmospheric Measurement Techniques, 2018_

## Short Comment (SC1) · 16 Jun 2018

I was very intrigued to see this paper, as I believe that the research community needs to make *many* profiles with sensor like these to truly help evaluate the accuracy of radiative heating rate profiles in the Arctic (and other atmospheres), as traditionally these profiles have only been computed using atmospheric and cloud properties and radiative transfer models. Thus, I am very supportive of this work.

However, the last sentence of the abstract is misleading. The LW radiative cooling seen at the top of a stratiform cloud in the Arctic can be quite variable, as it depends on the liquid water path (LWP) of the cloud, the distribution of that liquid water vertically, whether there is ice in the cloud also (i.e., if it is mixed-phase or not) and if there are

other cloud layers above the one being sampled. These issues have been discussed in papers such as Turner et al. J. App. Meteor. Clim. 2018. I would encourage the authors to broaden there discussion in section 4 to include caveats such as this, and to indicate how remote sensors or additional instruments on the tethered balloon system will be needed to address these concerns.

However, it is also important to understand the uncertainties in this tethered measurement approach. Would the authors be able to provide uncertainty estimates for both their flux and radiative heating rate values? This seems to be a critical piece of information to know how far these observations can be trusted.

Thanks.
* * *

---

## Short Comment (SC2) · 22 Jun 2018

Thanks for the comments and encouraging suggestions !

The discussion on cloud radiative properties here is based on a sequence of four subsequent profiles only. All results presented provide a characterisation of this individual situation. It should be noted that tethered ballooning cannot be performed to obtain multi-year datasets (as described in the study you are referring to), but for case studies.

More data on microphysical properties of the sensed clouds would be beneficial to get more realistic simulation results. Recently, Alfred-Wegener-Institute started operation of cloud radar at Ny Alesund site.

[Figure]

We managed to keep uncertainties in the fluxes as small as possible to get realistic heating rates by selecting best hardware, i.e. using instruments operated in the BSRN context. Both instrument types - pyranometer and pyrgeometer - are linked to the world standard group/world infrared standard group by its calibration procedures. Concerning the misalignment during flight it can be said that the mean deviation in sun zenith (calculated minus observed) is about $1.8°$ for the four profiles. Thus, misalignment can be neglected for the diffuse radiative fluxes. For shortwave downward, facing sun zenith of about $60°$, we get a correction term of 17 W/m2. Hence, after applying the correction it is assumed to get an uncertainty only marginally enhanced w.r.t. near surface observations. Parallel observations with mast-mounted fast sensors would help to further consolidate that.

---

## Referee Comment (RC1) · Anonymous Referee #1 · 18 Jul 2018

**1   General remarks**

The manuscript introduces a new tethered balloon sonde measuring solar and terrestrial up- and downward radiative flux densities.  The sensors package is described, measurement uncertainties are briefly discussed and correction methods are presented. Some limited number of observations is used to illustrate the potential of this observation platform.

The new instrument package, the measured profiles of radiative flux densities and derived heating rates are within the scope of AMT and might contribute to improve our understanding of radiative processes in the lower clear sky and cloudy atmosphere. However, that manuscript is of poor quality which I will try to describe in detail below

[Figure]

and, therefore, does not exhaust its full potential.

The methods applied to process the measurements are oversimplified and do not use state of the art methods, nor state of the art methods are compared or discussed. The methods applied in the manuscript and other details of data processing, simulations are not or inaccurately specified in many parts of the manuscript making it impossible to follow or reproduce the data processing and calculations. The performance of the instruments is not well characterized. Literature used for discussion is often not state of the art. Final uncertainties of the measurements are not given. Figures, style of presentation (structure, English, use of abbreviations, symbols, etc.), are below the quality level of AMT.

Therefore, the manuscript is far from a status that it can be published in a high quality journal like AMT. Because I like the measurement approach and see the potential of such observations, I only can recommend to revise the manuscript in all matters suggested below.

Below, I compiled a list of comments which have to be considered in a revised version of the paper. There might be some contradictory statements which result from my misinterpretation of the text when first reading the not well organized manuscript. I am sure the authors will know how to weight in such cases and how to improve the text to avoid misinterpretations by other readers.

**2 Major comments**

**Tilt correction**

There exist references which quantify the errors of a sensor tilt in high detail. e.g. Bannehr and Schwiesow (1993), Wendisch et al. (2001). Those are not included in the discussion of the manuscript but may easily point to the problem and magnitude of

sensor tilt.

Also common correction methods frequently applied to airborne radiation measurements are not compared to the method presented here, nor are these methods discussed (e.g., Bannehr and Schwiesow, 1993). Instead, a rather simplified approach is used to correct for the sensor tilt. This approach is not well explained in the manuscript. Details are missing and assumptions like neglecting diffuse radiation or assuming the diffuse radiation to be height independent are questionable. Additionally, Figure 4 seems to have a bug. At least it show larger tilt errors for high Sun (zenith position) which does not follow theory. So I had problems following the method.

Only one example of measurements at a different location is shown, which certainly does not cover all possible conditions for which the correction needs to be applied. In the end it also was stated, that simulated downward irradiance were used for the analysis of the examples. Finally, I completely lost track of what had been done with the measurements to correct for the sensor tilt and how large the remaining uncertainty is after the correct. Some detailed comments and questions are here and in the list of comments below.

**L149:** Neglecting uncertainties of a sensor tilt for terrestrial radiation and upward solar radiation might be not acceptable. There should be uncertainties for $10°$, which at least need to be quantified. Or a reference needs to be given discussion this problem. Can you estimate what error a $10°$ tilt would cause if a bright surface and a dark sky would be assumed. Should be possible to calculate from geometry.

**L156, Figure 2:** For what SZA this example holds? The effect of $5°$ tilt will be different if the Sun is in zenith. Therefore, I do not understand why this should be shown with a single measurement case. It is possible to calculate the effect from geometry for all possible geometries.

**L162:** Calculation of individual solar zenith angle: How this transformation is calculated? Provide the equations or a reference. However, it will not help to calculate the

individual solar zenith angle if you want to analyse vertical profiles and compare two measurements with different tilt. I do not understand this approach.

**L167-169:** This assumption is highly risky in my point of view. Especially, if vertical profiles are to be analysed. Direct/diffuse fraction will change in dependence of altitude. Therefore, I recommend to use the analytical equations that can be used to correct for the sensor tilt. Only the direct/diffuse fraction needs to be known. This can be estimated by radiative transfer simulations.

**L174:** Which day? Which solar zenith angle? Is this example comparable to Arctic observations? Are the conclusions transferable to the Arctic?

**Figure 4:** I do not understand, why the misalignment error should be larger for low solar zenith angles (high Sun)? Cosine law tells, that changes for solar zenith angles of $0°$ (sun in zenith) are way less than changes of low sun, e.g., solar zenith angles of $80°$.

$$\cos(0°) \cdot 1000\,\mathrm{Wm}^{-2} - \cos(3°) \cdot 1000\,\mathrm{Wm}^{-2} = 1\,\mathrm{Wm}^{-2}$$

$$\cos(77°) \cdot 1000\,\mathrm{Wm}^{-2} - \cos(80°) \cdot 1000\,\mathrm{Wm}^{-2} = 51\,\mathrm{Wm}^{-2}$$

To conclude: This correction scheme needs to be explained and tested in much more detail. In order to make the study shown in the manuscript more relevant, the correction approach needs to be explored for a general application to measurements in all potential conditions. Radiative transfer simulations are needed to support the approach. Two measurements at one location on a single can not be used to derive a parametrization for such a correction. Alternatively, the authors may apply common methods to correct for the sensor tilt.

[Figure]

**"Lack of detail and discussion with respect to state of the art"**

In almost all parts of the manuscript specifications of methods, instrument performance, etc. is missing. Day, time, solar zenith angle of observations is missing. Known methods reported in literature are not discussed. Therefore, the manuscript does not represent the state of the art of solar and terrestrial radiation measurements and analysis of vertical profiles of heating rates. Just to mention some references which need to be considered by the authors:

Bannehr, L. and Schwiesow, R. , 1993: A technique to account for the misalignment of pyranometers installed on aircraft, J. Atmos. Oceanic Technol., 10, 774– 777.

Freese, D. and Kottmeier, C.: Radiation exchange between stratus clouds and polar marine surfaces, Bound.-Lay. Meteorol., 87, 331–356, 1998.

Duda, D. P., Stephens, G. L., and Cox, S. K.: Microphysical and radiative properties of marine stratocumulus from tethered balloon measurements, J. Appl. Meteo., 30, 170-186, 1991.

Bucholtz, A., Hlavka, D. L., McGill, M. J., Schmidt, K. S., Pilewskie, P., Davis, S. M., Reid, E. A., and Walker, A. L.: Directly measured heating rates of a tropical subvisible cirrus cloud RID C-9570-2011, J. Geophys. Res., 115, D00J09, doi:10.1029/2009JD013128, 2010.

Saunders, R. W., Brogniez, G., Buriez, J. C., Meerkötter, R., and Wendling, P.: A comparison of measured and modeled broadband fluxes from aircraft data during the ICE'89 field experiment, J. Atmos. Ocean. Tech., 9, 391–406, 1992.

Gerber, H., S. P. Malinowski, A. Bucholtz, and T. Thorsen, 2014: Radiative cooling of stratocumulus. 14th Conf. on Atmospheric Radiation, Boston, MA, Amer. Meteor. Soc., 9.3. [Available online at https://ams.confex.com/ams/14CLOUD14ATRAD/webprogram/Paper248451.html.]

Curry, J. A. and Herman, G. F.: Infrared radiative properties of summertime Arctic stratus clouds, J. Climate Appl. Meteor., 24, 525–538, doi:10.1175/1520-0450(1985)024<0525:IRPOSA>2.0.CO;2, 1985.

Wendisch, M., D. Müller, D. Schell, and J. Heintzenberg, 2001: An airborne spectral albedometer with active horizontal stabilization. J. Atmos. Ocean. Tech., 18, 1856-1866.

**Scientific presentation**

The manuscript does not reach the standard of scientific writing required for AMT. Partly due to reasons given above, but also due to some general shortcoming.

Figures: Figures are of low standard. Labels are not precise and label boxes displaced from the main figure. Lines are partly not distinguishable. Figure captions do not explain, what is actually shown. Axis labels are incomplete or abbreviations not introduced before. Parametrizations are given in the figure instead of as equation within the text.

Quantities, units: Quantities are not used in a correct way, e.g. flux vs. irradiance (radiative flux density). Many quantities are given as abbreviations. Units are not written with exponentials. $°$ written as "deg". Date and time format is not consistent.

**3 List of comments**

**L15:** "four times net" There are only two net radiative flux densities, solar and terrestrial.

**L20:** For which altitude the albedo from balloon observations is given? Why there are two values for surface albedo and albedo in flight level?

**L20:** Rather present albedo in [0,1]. Use of percentage is quite unusual.

**L21:** This sentence reads like a general finding, but this maximum is likely only valid for the specific location of the measurements (horizontal surface inhomogeneity).

**L38:** Which decades?

**L39:** Reference for IPCC report: Why 2007? There are new versions of the report. Does this statement still hold for the latest IPCC report?

**L50:** Stick to one unit. In the introduction $k\,day^{-1}$ were used. My feeling is, that $K\,h^{-1}$ is more useful for the low level clouds analyzed in this study.

**L58 and whole manuscript:** "Irradiation profiles", "fluxes": Be more precise or at least consistent with the radiation quantity measured. It is "radiative flux density" or "irradiance" to be correct. "radiative flux" as stated in the title already is incorrect. Unit of "radiative flux" is W while "radiative flux density" has a unit of $W\,m^{-2}$.

**L62-65:** Reference to Siebert et al. (2003) does not fit here, when discussing radiation measurements as there had been no radiation measurements in Siebert et al. (2003). Rather use this reference as a general application of balloon borne measurements.

**L66:** Reference to Philipona et al. (2012) and Kräuchi and Philipona (2016). What is

the difference of these studies to your measurement setup? Give some more details about what has been done in these references.

**L68:** A sonic is no satellite-based remote sensing technique. Also sonic is a ground-based observation and therefore already an in situ instrument. Rather discuss how ground-based and satellite remote sensing derived solar and terrestrial radiation profiles (general approach, remote sensing of atmospheric state and calculation of radiation based on radiative transfer simulation). What are the limitation compared to balloon borne observations?

**L88:** First introduce the sensor type, then give uncertainties.

**L90:** WMO (2008): This is not the latest edition of the WMO guide. And the reference does not tell about the uncertainty of the sensor used here. It only defines quality standards. Give the specific uncertainty of your instrument.

**L93:** "reduced sensitivity": specify!

**L95:** "secondary standard": Rather give numbers of uncertainties here.

**L101:** Radiation shields have been removed: How does this affect the uncertainties? Still "secondary standard"?

**L106:** Data loggers: Specify sampling frequency! Specify accuracy of the tilt sensor!

**L108:** "from 1 second to 1 minute": What was finally uses?

**L110:** CM11: I don't care about the CM11. Give range of CM22 used in your study.

**L114:** What is B? If that is a quantity then use italic letter.

**L115:** What is the ascent rate? Maybe restructure your manuscript and first introduce the balloon system then the sensors.

**L116:** I don't understand. What would it help to place the sensor into ambient conditions, when the ambient temperature will change during the ascent?

**L124:** Filled with helium or hydrogen? What is the maximum lifting weight?

**L128:** "several sondes": Several? Are these mounted at the same time in different altitudes?

**L131:** What if time drift during the flight is strong? Simply state the uncertainty in time difference you assume for your system.

**L139:** "minimize mixing of temporal and spatial (vertical) change". Spatial vertical changes are what you are about to measure. So these should not be minimized.

**L153:** "high potential errors that need to be corrected" Why you can draw this conclusion? First provide the analysis which supports this conclusion.

**L159:** Give specification of inclinometer!

**L162:** How the wind vane of the radiation package can measure the wind direction? Is there a reading?

**L164:** Give a reference or justify why no correction is needed in overcast conditions.

**L174:** What steps?

**L177:** What is "quite similar"? To compare the slope of the plot, what is what you are looking for, the data first needs to be normalized.

**L194:** There are references which discuss that it is necessary to correct the measurements for the response time of the sensors in order to properly correct for the tilt of the solar irradiance measurements, e.g., Freese and Kottmeier (1998) or Ehrlich and Wendisch (2015).

**L192:** What level? Altitude $z$ should be used as vertical coordinate instead of an index $i$.

**L206, Eq. 3:** What does the $\delta$ mean? Difference or derivative? If you use discrete altitudes (levels) you need to write the equation also for discrete levels.

**L261-267:** This section is rather something for the introduction.

**L268:** Of course it is. Radiative transfer processed always determine the solar irradiance, also in a 3D world, and for a large sensor footprint.

**L274:** This statement is too trivial. Remove.

**L279:** Why using simulations for downward irradiance? Wasn't downward irradiance measured? The discussion on the tilt correction suggested that there are measurements. Use measurements!

**L280:** "absence of insolation": Why analyzing solar radiation, if there is no solar radiation at all (no insolation = night)?

**L287:** Can you explain the maximum?

**L292:** Ceilometer might not be the best indicator of clouds disturbing the direct solar radiation. Clouds can be in front of the Sun, but not registered by the ceilometer in zenith.

**L302:** The horizontal drift will have an effect on the profiles of albedo presented before! A changing drift in different altitudes may bias the profiles or even produce artificial changes in the radiation profiles. How large was the effect for the measurements of albedo profiles?

**L309:** What is the benefit of measuring long time series of albedo? This is not clear from this section. Only the mean values are used to explain the effect of the balloon drift. No time series are needed for this effect.

**L319:** What is "May13 1"?

**L319 and Figure 10:** Introduce figure and what is plotted properly not only as a hint in some brackets.

**Figure 10:** Why showing potential temperature? Emission (Planck's law) depends

on air temperature. So to illustrate higher emission by higher altitudes, the real air temperature is needed, not the potential temperature.

**L323-325:** Why? If there is no impact on net radiation, where you will have a warming of the atmosphere leading to Arctic amplification? Does it has an effect on the surface?

**L327:** What do you mean with radiative effect? Radiative forcing of the cloud at surface? Specify!

**L327:** How do you compare cloudy case with clear sky case? Measurements of the same day, time? Same atmosphere profile? In best case, this needs to fixed (identical) in order to compare the impact of clouds on the radiation profile.

**L336:** UTC time? Use 24h format.

**L349:** "passing": This is not unambiguous. Passed from which side? Ascend or descend?

**L348-365:** Figure 13 is discussed in both sections. I suggest to reorganize this part in order to have the introduction and discussion in one section only.

**L360:** What signal? Irradiance is zero?

**L364:** There is no figure showing an albedo profile.

**L366:** "total net effect": What effect? Be more precise!

**L371:** The cloud layer indicated by the gray area in the figure is below the maximum of cooling rates. But cooling rates should be inside the cloud not above. Needs to be discussed and compared to literature.

**L402:** Repetition. Was already mentioned when introducing the table.

**L407:** The heating rate profile suggest, that cloud top was higher. I suggest to add the simulated profiles of heating rates into Figure 13.

**L409:** The different cloud top and cloud base altitudes for the single profiles need to

be indicated in Figure 13 where only one altitude is shown.

**L410:** Cloud base warming strongly depends on the surface temperature (difference to cloud base temperature). What is used in the model, what was observed?

**L413:** How heating rates are derived from the model simulations? Heating rates strongly depend on the layer thickness considered in the calculations. On what vertical grid the simulations were performed and analyzed? To compare with the observations the same layer needs to be analyzed. Was this done?

**L418:** I suggest to use $W\,m^{-2}$ instead of relative values for the comparison.

**L431-432:** The potential still needs to be shown! This is only speculative. I suggest to remove such statements.

**Figure 1:** Indicate which sensor is CM22 and which CG4. Where is the inclinometer mounted?

**Figure 2:** Labels: "Shortwave downward..." what? Write $W\,m^{-2}$. What is Sunzen? 130305 is this 13 March 2005 or 3 May 2013 or 5 March 2013? What does this equation in the label stands for? The equation does not use symbols for quantities introduced before. What is sz2? What is sz? What units do the coefficients of the equation own?

**Figure 3:** Similar to Figure 2: Incorrect quantities. What is GAP? What is GAP Sel, ZUG,...? Do not place parametrizations in the legend. Figure caption: The plot does not show "data fitting" it shows downward solar irradiance observed for different solar zenith angles.

**Figure 4:** Measurements and parametrization are only shown for 35-75° solar zenith angle. How you can calculate for 0° and 80°? Parametrization is not valid for these angles.

**Figure 4:** Label: What is "Diff" ? Give correct quantity. Caption is not stating what actually is shown.

**Figure 5:** Starting at low altitude over a bright snow surface I do not understand, how the reflected radiation can increase over the first 200 m. This increase looks like you started over a locally darker surface. How these profiles compare to the BSRN data at the surface?

**Figure 6:** Caption: What do you mean with range? Min and max?

**Figure 7:** Figure is not discussed in text. Therefore, there is no need to snow the time series of T and r.H. What is UT? "deg" as unit.

**Figure 8:** Merge with figure 9. This will allow to compare wind speed and direction with albedo.

**Figure 10:** Two lines are undistinguishable. What is LWD?

**Figure 11:** Give day, time of observations.

**Figure 13:** Give solar zenith angle! Differences between the profiles might result a from different solar position.

**Figure 14:** Label all symbols. What is total what is terrestrial cooling? What is LHR?

**Figure 14:** How large are the uncertainties of derived heating rates? Uncertainties etc. need to be given to illustrate the potential of the measurements.

**4  References used in the review**

Freese, D. and Kottmeier, C.: Radiation exchange between stratus clouds and polar marine surfaces, Bound.-Lay. Meteorol., 87, 331–356, 1998.

Ehrlich, A. and Wendisch, M., Reconstruction of high-resolution time series from slow-response broadband terrestrial irradiance measurements by deconvolution,

[Figure]

Atmos. Meas. Tech., 8 , 3671-3684, doi:10.5194/amt-8-3671-2015., 2015

Wendisch, M., D. Müller, D. Schell, and J. Heintzenberg, 2001: An airborne spectral albedometer with active horizontal stabilization. J. Atmos. Ocean. Tech., 18, 1856-1866.

Bannehr, L. and Schwiesow, R. , 1993: A technique to account for the misalignment of pyranometers installed on aircraft, J. Atmos. Oceanic Technol., 10, 774– 777.

---

## Author Comment (AC1) · 31 Jul 2018

Thanks for your comments. The approach to quantify the tilting chosen in this work refers to the Euler angles of the sensor, measured at the pyranometer pair body. Discrepancies between expected and measured sun zenith angle are causing a loss or a yield of the direct component of the total solar incoming radiation. The magnitude of the correction is to be taken from measured (and then fitted) near surface data of solar downward radiation as a function of sun zenith angle. It is assumed that the rate of change is invariant with height, at least at levels below 3000 m above sea level. An example of two closely located sites is given. I understand your point that these assumption would need further experimental or theoretical - by radiative transfer modelling - proof. Our conclusion here is to take this aspect out for further discussion on

tilting correction elsewhere and revise the manuscript according to your suggestion by integrating a well-documented approach for misalignment correction, discussed e.g. in Bannehr and Schwiesow, 1993 and Wendisch et.al. 2001. Of course that would imply a recalculation of part of the presented data. All issues in the list of comments will be regarded in a revised version, too.

---

## Referee Comment (RC2) · Anonymous Referee #2 · 17 Sep 2018

Review of: "In-situ sounding of radiation flux profiles through the Arctic lower troposphere"

Authors: R. Becker, M. Maturilli, R. Philipona, K. Behrens

General comments: The authors provide a look into recent efforts to make broadband radiation measurements from a tethered-balloon platform. In general, I believe this work to be of great value, as there is a substantial need to conduct more profiling of radiative properties throughout the lower atmosphere. In order for such measurements to truly be useful, it is important to collect profiles over a wide-variety of conditions, and development of instrumentation that can be deployed on a semi-regular (albeit weather condition dependent) manner is useful. Therefore, I applaud the authors in taking steps toward such an ability.

[Figure]

Having said this, there are several issues with the current manuscript that require further attention before it is ready to be published. I outline these below, doing my best to divide them between "major" and "minor" issues requiring attention. In general, I believe that this manuscript requires major revisions before it can be considered for final publication.

Major Issues:

- The authors use inconsistent terminology. It is important to distinguish between "flux" and "flux density" or "irradiance". Please check your units and usage of these terms and make the appropriate changes (including to the title).

- The introduction goes directly from towers to balloons and kites, completely ignoring the work that has been conducted using manned and unmanned research aircraft. I recommend that the authors dig a little deeper into the history of aerial radiation measurements, as it will provide additional insight into several relevant topics, including tilt correction which I found to be discussed in insufficient detail.

- I found discussion of several topics to be lacking or incomplete. As the manuscript reads currently, it seems like a gathering of thoughts more than a thorough scientific paper. For example:

o Section 2.2: Synchronization of logging rates was deemed to not be a critical issue because of the slower response of the radiation sensors. This comment doesn't make sense to me. Ultimately, being able to match up the radiation sensors with the sensors measuring platform attitude is still critical for tilt correction, developing profiles and more. More information is needed to justify this statement and more details on the logging system would be helpful.

o There is insufficient discussion on the tilt correction algorithm. If I understand correctly, Figure 2 shows the error associated with a 5-degree change in the solar zenith angle from ground-based measurements, but this tells us very little about the error

associated with a sensor that is misaligned by 5 degrees. That is because an actual change in solar zenith angle also changes the pathlength of the sun through the atmosphere, which is part of why there is a chance in the radiative flux density. However, changing the sensor tilt angle at a given path length has a different effect. Additionally, the example provided, while for clear sky, only considers one atmospheric state, and does not account for what happens when there is more or less water vapor present during that shift in sensor tilt. A more rigorous analysis of what the true impact of sensor tilt is needed. Additionally, a much more thorough overview of the tilt-correction algorithms applied is required, along with (particularly for AMT) a more detailed discussion of the instrumentation used to determine sensor attitude (pitch, roll, yaw). For example, is the yaw from a magnetometer? If so, what is the impact on the measurement at high latitudes? Was the magnetometer calibrated to the local declination angle before flight? What are the expected uncertainties associated with the inclinometer in a static (i.e. non-moving) condition? To me, the "calculations" section should really focus on these items, not the much more trivial equations related to radiative flux density.

- In my opinion, the radiative transfer simulations, are inadequate. For example, the microphysical properties of the clouds are assumed to be those reported by Curry and Ebert. There are many other studies that have investigated cloud microphysics in Arctic stratiform clouds in many different locations and seasons. While it is challenging to say which of these studies are most representative of the conditions observed in this case, at the very least some level of sensitivity study should be completed to evaluate how much the microphysical parameters impact the calculated radiative profiles.

- Additionally, the radiative transfer simulations offer an opportunity to conduct some sensitivity studies to parameters implicated in this study. For example, could the authors look at the impact of effective surface albedo and evaluate to what extent this impacts the profile? This would help to assess whether the differences between the measured quantities at the surface or aloft are realistic. Profiles over a range of quantities could be compared directly to the measured quantities in one of the figures.

- In general, the amount of discussion included in the manuscript is lacking, and there are numerous unclear connections made in the text. For example:

o There is reference to the cloudy conditions having higher measured albedo than the clear conditions. I assume this is due to multiple reflections, but there is no discussion on it.

o Multiple times, the variability in surface conditions and increased visibility of this variability is mentioned as the reasoning behind seeing lower surface albedo in the tethered balloon measurements than what is observed at the surface, but there is no discussion on how this is verified. For example, small errors associated with the tilt correction or attitude estimation could also result in increased downwelling irradiance, which would reduce albedo. More detail is required.

o Section 3.3: several comments on the radiative forcing are made, but it is not clear whether these are meant to be generalizations, or just for this specific case. For example, the comments on longwave flux at the end of the first paragraph in the section.

o Line 351: "only about half": Half of what? I see drop in the LWD of approximately 140 W/m2, and a drop in the LWU of only 50 W/m2. This doesn't seem like half, but maybe I'm misunderstanding. More detail/discussion is needed.

o There are several comments about something happening as the balloon passes through cloud top, or cloud base. However, there is generally no indication of which direction the instruments are moving during this transition. Is this from within the cloud to outside of the cloud? Or vice versa? Please be clear about these transitions in the text so that the reader doesn't have to guess at what you mean.

o Line 436: Weak relative to what?

o Line 437: Stronger relative to what?

- There is no discussion on sensor riming within a supercooled cloud layer. I presume that the sensors are not heated, based on the power required. How do the authors

know that riming is not a problem within cloud?

- There is limited discussion on the impact of assuming 1D radiative transfer (vs. 3D) in assessing differences between the simulated radiation and the observed values. There are likely to be implications, especially at a coastal boundary with multiple surface albedos.

- There are no estimates of the uncertainty of these measurements. Ultimately, these are critical for evaluating their value.

- The figures need to be more clearly explained in the captions. Line types and colors should be clearly and consistently explained in the captions. Additionally, maybe I missed it, but what does the sigma represent in the captions for figures 7, 8, and 9?

Minor Issues:

- I believe that "Key and Schwaiger" should be Key and Schweiger. Please check the spelling.

- Section 3.2, second paragraph: Flat terrain or not, it's the radiative transfer processes that control the measured radiative flux density. Just because the terrain is more complicated, doesn't mean that it's not radiative transfer impacting the measurements...

- A satellite image/map of the flight area would be useful. The authors could even include some range rings indicating the position of the balloon at a given altitude, assuming a tether tilt angle.

- Was the instrumentation directly below the balloon? Or how far below the balloon were the instruments mounted?

- There are some limited grammatical issues, and I recommend that the manuscript be reviewed for spelling and grammar.

---

## Author Comment (AC2) · 12 Oct 2018

The authors wish to thank the reviewer for his critical comments that provided a constructive guidance to improve the manuscript. Please find our detailed response below.

Ref: The authors use inconsistent terminology. It is important to distinguish between "flux" and "flux density" or "irradiance". Please check your units and usage of these terms and make the appropriate changes (including to the title).

-> The terminology will be corrected/streamlined.

Ref: The introduction goes directly from towers to balloons and kites, completely ignoring the work that has been conducted using manned and unmanned research aircraft. I recommend that the authors dig a little deeper into the history of aerial radiation mea-

surements, as it will provide additional insight into several relevant topics, including tilt correction which I found to be discussed in insufficient detail.

-> The revised manuscript version shall discuss results of research aircraft activities including the tilt correction issue.

Ref: I found discussion of several topics to be lacking or incomplete. As the manuscript reads currently, it seems like a gathering of thoughts more than a thorough scientific paper. For example: Section 2.2: Synchronization of logging rates was deemed to not be a critical issue because of the slower response of the radiation sensors. This comment doesn't make sense to me. Ultimately, being able to match up the radiation sensors with the sensors measuring platform attitude is still critical for tilt correction, developing profiles and more. More information is needed to justify this statement and more details on the logging system would be helpful.

-> all sensors and receiving systems are synchronized whether they are fast or slow. We define 'no critical issue' that we ignore the sub-second range.

Ref: There is insufficient discussion on the tilt correction algorithm. If I understand correctly, Figure 2 shows the error associated with a 5-degree change in the solar zenith angle from ground-based measurements, but this tells us very little about the error associated with a sensor that is misaligned by 5 degrees. That is because an actual change in solar zenith angle also changes the pathlength of the sun through the atmosphere, which is part of why there is a chance in the radiative flux density. However, changing the sensor tilt angle at a given path length has a different effect. Additionally, the example provided, while for clear sky, only considers one atmospheric state, and does not account for what happens when there is more or less water vapor present during that shift in sensor tilt. A more rigorous analysis of what the true impact of sensor tilt is needed. Additionally, a much more thorough overview of the tilt-correction algorithms applied is required, along with (particularly for AMT) a more detailed discussion of the instrumentation used to determine sensor attitude (pitch, roll, yaw). For example, is the yaw from a magnetometer? If so, what is the impact on the measurement at high latitudes? Was the magnetometer calibrated to the local declination angle before flight? What are the expected uncertainties associated with the inclinometer in a static (i.e. non-moving) condition? To me, the calculations" section should really focus on these items, not the much more trivial equations related to radiative flux density

-> We see your point, it is one of the major issues of the other referee, too. Rethinking this aspect, the idea to make of use surface based measured data shall be rejected because of too much uncertainty entering. We'll refer to the correction equation provided by Bannehr & Schwiesow 1992 further on. - The magnetometer of Vaisala meteorological sonde is calibrated onsite before flight. It is assumed that the error in wind direction caused by the deviation between magnetic and geographic north is lower than the uncertainty of measurements itself (5 deg.). This error needs to be regarded in an uncertainty assessment.

Ref: In my opinion, the radiative transfer simulations, are inadequate. For example, the microphysical properties of the clouds are assumed to be those reported by Curry and Ebert. There are many other studies that have investigated cloud microphysics in Arctic stratiform clouds in many different locations and seasons. While it is challenging to say which of these studies are most representative of the conditions observed in this case, at the very least some level of sensitivity study should be completed to evaluate how much the microphysical parameters impact the calculated radiative profiles. Additionally, the radiative transfer simulations offer an opportunity to conduct some sensitivity studies to parameters implicated in this study. For example, could the authors look at the impact of effective surface albedo and evaluate to what extent this impacts the profile? This would help to assess whether the differences between the measured quantities at the surface or aloft are realistic. Profiles over a range of quantities could be compared directly to the measured quantities in one of the figures.

-> Due to the lack of instantaneous observations the input parameters were selected according to Curry & Ebert (1992). A sensitivity study to estimate the impact of these

parameters was considered to be beneficial, the same with effective surface albedo (see below)

Ref: There is reference to the cloudy conditions having higher measured albedo than the clear conditions. I assume this is due to multiple reflections, but there is no discussion on it

-> Albedo cloudy/clear-sky: on average the diffuse albedo profiles show higher values than in clear-sky conditions. From clear-sky to overcast, the drop in shortwave downward radiation is stronger than in shortwave upward. (Figure 6)

Ref: Multiple times, the variability in surface conditions and increased visibility of this variability is mentioned as the reasoning behind seeing lower surface albedo in the tethered balloon measurements than what is observed at the surface, but there is no discussion on how this is verified. For example, small errors associated with the tilt correction or attitude estimation could also result in increased downwelling irradiance, which would reduce albedo. More detail is required.

-> The issue 'variability in surface conditions' implies two main aspects: in case of changing cloudiness conditions like 'cloud-shadow on surface and sun-exposed lifted pyranometer' and vice-versa may happen. This is expected to result in a peak (a drop) in albedo, respectively. You'll never get this with near-surface observations. The other source of surface variability regards surface albedo. Near-surface observation is staring at snow surface whereas lifted radiometers sense a mixed scene composed by snow-covered land and ice-free water. This aspect could be investigated further in the RTM section.

Ref: Section 3.3: several comments on the radiative forcing are made, but it is not clear whether these are meant to be generalizations, or just for this specific case. For example, the comments on longwave flux at the end of the first paragraph in the section. o Line 351: "only about half": Half of what? I see drop in the LWD of approximately 140 W/m2, and a drop in the LWU of only 50 W/m2. This doesn't seem like half, but

maybe I'm misunderstanding. More detail/discussion is needed

-> Sect 3.3/L351: discussion regards measurements taken on May 12, 2015, only. Above cloud LWD is about half of LWU, below and in the cloud it differs by only 10 to 20 W/m2. To be reformulated

Ref: There are several comments about something happening as the balloon passes through cloud top, or cloud base. However, there is generally no indication of which direction the instruments are moving during this transition. Is this from within the cloud to outside of the cloud? Or vice versa? Please be clear about these transitions in the text so that the reader doesn't have to guess at what you mean

-> Transitions shall be marked, preferably in section 4.2. Concerning gradients it doesn't matter

Ref: Line 436: Weak relative to what? Line 437: Stronger relative to what ?

-> L436/437: compared to each other

Ref: There is no discussion on sensor riming within a supercooled cloud layer. I presume that the sensors are not heated, based on the power required. How do the authors know that riming is not a problem within cloud?

-> Riming can be a problem, and it should be discussed here shortly. We did not observe riming on May 11th and 12th when we got the desired shallow low-level clouds. It takes about 6 minutes for the instrument to descent from the cloud base to the surface, and ice or riming was found neither on the domes nor the instrument bodies. Considered that there was no insolation supporting melting/sublimation we would have seen it then.

Ref: There is limited discussion on the impact of assuming 1D radiative transfer (vs. 3D) in assessing differences between the simulated radiation and the observed values. There are likely to be implications, especially at a coastal boundary with multiple surface albedos

-> 1D vs 3D simulations: possible implications should be mentioned

Ref: There are no estimates of the uncertainty of these measurements. Ultimately, these are critical for evaluating their value.

-> A solid uncertainty estimation is needed

Ref: The figures need to be more clearly explained in the captions. Line types and colors should be clearly and consistently explained in the captions. Additionally, maybe I missed it, but what does the sigma represent in the captions for figures 7, 8, and 9?

-> Better explanation of figures. Sigma in Fig 7,8,9 means standard deviation of geometrical height of the balloon (sensor)

---

## Author Comment (AC3) · 1 Nov 2018

The authors wish to thank the reviewer for his/her critical comments that provided a constructive guidance to improve the manuscript. Please find our detailed response to the major comments below.

**Tilt correction**

There exist references which quantify the errors of a sensor tilt in high detail. e.g. Bannehr and Schwiesow (1993), Wendisch et al. (2001). Those are not included in the discussion of the manuscript but may easily point to the problem and magnitude of sensor tilt. Also common correction methods frequently applied to airborne radiation measurements are not compared to the method presented here, nor are these methods discussed (e.g., Bannehr and Schwiesow, 1993). Instead, a rather simplified approach is used to correct for the sensor tilt. This approach is not well explained in the manuscript. Details are missing and assumptions like neglecting diffuse radiation or assuming the diffuse radiation to be height independent are questionable. Additionally, Figure 4 seems to have a bug. At least it show larger tilt errors for high Sun (zenith position) which does not follow theory. So I had problems following the method.

➔ *Figure 4 shows the error in shortwave downward radiation as a function of sun zenith angle. It is delineated from the data fits displayed in figure 3 and further on it is assumed that based on such fits individual (scene dependent) functions can be obtained. Please consider that figure 4 is displaying absolute units in contradiction to the reference Bannehr & Schwiesow 1992, using percentage error. Recalculating data for figure 4 to percentage error would show an expected behavior: higher errors at low sun altitudes.*

Only one example of measurements at a different location is shown, which certainly does not cover all possible conditions for which the correction needs to be applied. In the end it also was stated, that simulated downward irradiance were used for the analysis of the examples. Finally, I completely lost track of what had been done with the measurements to correct for the sensor tilt and how large the remaining uncertainty is after the correct. Some detailed comments and questions are here and in the list of comments below.

**L149:** Neglecting uncertainties of a sensor tilt for terrestrial radiation and upward solar radiation might be not acceptable. There should be uncertainties for 10°, which at least need to be quantified. Or a reference needs to be given discussion this problem. Can you estimate what error a 10° tilt would cause if a bright surface and a dark sky would be assumed. Should be possible to calculate from geometry.

➔ *Mean misalignment for the cloud flights on May 12, 2015, is 1.8°. Saunders et.al. (1992) is stating that 'in diffuse radiation the attitude of the pyranometer is not critical since the measured flux does not vary significantly with the attitude of the pyranometer'. This is given with the cloud flights. It is agreed that bright surface/dark sky aspect might be an issue with the albedo flights. There are two things to consider, first to keep out data points obtained in large misalignments (>10°), second the topography around Ny Alesund. It is not a sky flight, we do not exceed 2 km height – tilting the sensor will rather add minor contributions from the surrounding low mountains (thus, snow) and atmospheric scattering.*

**L156, Figure 2:** For what SZA this example holds? The effect of 5° tilt will be different if the Sun is in zenith. Therefore, I do not understand why this should be shown with a single measurement case. It is possible to calculate the effect from geometry for all possible geometries.

➔ *Fig 2 shows exemplary, based on measurement data obtained in Lindenberg on March 05, 2013, that a misalignment of the senor, causing apparent sun zenith change from 65° to 70°, would lead to a loss in shortwave downward of 96 W/m2. It is agreed, that a more generalized statement, covering the range of expected SZA, matches better.*

**L162:** Calculation of individual solar zenith angle: How this transformation is calculated? Provide the equations or a reference. However, it will not help to calculate the individual solar zenith angle if you want to analyse vertical profiles and compare two measurements with different tilt. I do not understand this approach.

➔ *Equations are taken from https://en.wikipedia.org/wiki/Euler_angles. SZA and sun azimuth for a horizontal plane are calculated using 'zenith'-function come along with libradtran-package (Mayer*

*and Kylling, 2005, should be cited too and added to the references). Polar coordinates are converted to Cartesian using*

*x=r\*sin(SZA)\*cos(azimuth)*
*y=r\*sin(SZA)\*sin(azimuth)*
*z=r\*cos(SZA)*

*Measured yaw angle (radiation sensors are mounted on a wind vane), pitch and roll from inclinometer provide the true 3d state of the sensor. Building the rotation matrice*

*rotmat = [ [cos(theta)\*cos(psi), cos(theta)\*sin(psi), -sin(theta)],$*
  *[sin(phi)\*sin(theta)\*cos(psi)-cos(theta)\*sin(psi), sin(phi)\*sin(theta)\*sin(psi)+cos(phi)\*cos(psi), sin(phi)\*cos(theta)],$*
  *[cos(phi)\*sin(theta)\*cos(psi)+sin(phi)\*sin(psi), cos(phi)\*sin(theta)\*sin(psi)-sin(phi)\*cos(psi), cos(phi)\*cos(theta)] ]*

*and application on (x,y,z) gives (x',y',z') and converting back to polar coordinates the apparent SZA'. Following the idea introduced here the difference to SZA corresponds to a yield/loss in shortwave downward according to the daily data fit.*

**L167-169:** This assumption is highly risky in my point of view. Especially, if vertical profiles are to be analysed. Direct/diffuse fraction will change in dependence of altitude. Therefore, I recommend to use the analytical equations that can be used to correct for the sensor tilt. Only the direct/diffuse fraction needs to be known. This can be estimated by radiative transfer simulations.
➔ *Agreed.*

**L174:** Which day? Which solar zenith angle? Is this example comparable to Arctic observations? Are the conclusions transferable to the Arctic?
➔ *It covers the whole day of Jul 22, 2013. This approach is insensitive to changes in atmospheric composition during the day. This is more a general restriction than an Arctic issue.*

**Figure 4:** I do not understand, why the misalignment error should be larger for low solar zenith angles (high Sun)? Cosine law tells, that changes for solar zenith angles of 0_ (sun in zenith) are way less than changes of low sun, e.g., solar zenith angles of 80_.

$\cos(0\_) \cdot 1000 W m_{-2} - \cos(3\_) \cdot 1000 W m_{-2} = 1 W m_{-2}$
$\cos(77\_) \cdot 1000 W m_{-2} - \cos(80\_) \cdot 1000 W m_{-2} = 51 W m_{-2}$
To conclude: This correction scheme needs to be explained and tested in much more detail. In order to make the study shown in the manuscript more relevant, the correction approach needs to be explored for a general application to measurements in all potential conditions. Radiative transfer simulations are needed to support the approach. Two measurements at one location on a single can not be used to derive a parametrization for such a correction. Alternatively, the authors may apply common methods to correct for the sensor tilt.

➔ *Please see the first comment. After this discussion it is agreed that there are two strategies to go on concerning tilting: first is to more generalize the scheme introduced here (would be an own study), second is to apply a referenced scheme. We'll choose second and follow recommendation given in your comment L167-169.*